# Evaluation of Antimicrobial Resistance of Different Phylogroups of *Escherichia coli* Isolates from Feces of Breeding and Laying Hens

**DOI:** 10.3390/antibiotics12010020

**Published:** 2022-12-23

**Authors:** Sandra Pais, Mariana Costa, Ana Rita Barata, Lígia Rodrigues, Isabel M. Afonso, Gonçalo Almeida

**Affiliations:** 1National Institute for Agricultural and Veterinary Research (INIAV, I.P.), Vairão, 4485-655 Vila do Conde, Portugal; 2Centre of Biological Engineering, Minho University (CEB), 4710-057 Braga, Portugal; 3Escola Superior Agrária de Ponte de Lima, Instituto Politécnico de Viana do Castelo, Refóios, 4990-706 Ponte de Lima, Portugal; 4Escola de Ciências da Vida e do Ambiente, Universidade de Trás-os-Montes e Alto Douro (UTAD), 5000-801 Vila Real, Portugal; 5LABBELS—Associate Laboratory, 4710-057 Braga, Portugal; 6CISAS—Centro de Investigação e Desenvolvimento em Sistemas Agroalimentares e Sustentabilidade-Escola Superior Agrária, Instituto Politécnico de Viana do Castelo, Rua Escola Industrial e Comercial de Nun’Álvares, n.º 34, 4900-347 Viana do Castelo, Portugal; 7Centre for Study in Animal Science (CECA-ICETA), Universidade do Porto, 4050-083 Porto, Portugal

**Keywords:** *Escherichia coli*, breeding hens, egg laying hens, fecal samples, phylogroup, MDR, ESBL

## Abstract

Animal and food sources are seen as a potential transmission pathway of multi-drug resistance (MDR) micro-organisms to humans. *Escherichia. coli* is frequently used as an indicator of fecal contamination in the food industry and known as a reservoir of antimicrobial resistance genes (ARGs). Microbial contamination as a major outcome for the poultry and egg industry and is a serious public health problem. In the present study we performed the quantification of β-glucoronidase positive *E. coli* in 60 fecal samples of breeding and laying hens collected in Portugal in 2019. Phylogenetic and pathotypic characterization, antimicrobial susceptibility, and detection of resistant extended-spectrum β-lactamase (ESBL) genes were assessed. The phylogenetic and pathogenic characterization and detection of ESBL genes were assessed by real-time PCR and antimicrobial susceptibility was evaluated using the disk diffusion method. Overall, *E. coli* quantification was 6.03 log CFU/g in breeding hens and 6.02 log CFU/g in laying hens. The most frequent phylogroups were B1. None of the isolates was classified as diarrheagenic *E. coli* (DEC). In total, 57% of the isolates showed MDR and 3.8% were positive for ESBL. Our study highlights that consumers may be exposed to MDR *E. coli*, presenting a major hazard to food safety and a risk to public health.

## 1. Introduction

Animal food products, such as eggs, meat, and milk, are abundant in proteins essential for the body’s maintenance, repair, and growth [1]. Poultry is among the most reported carriers of foodborne pathogens [2]. Hughes et al. [3] also asserted that poultry meat, red meat, and eggs are recognized as major vectors for the transmission of pathogens.

*E. coli* is a typical inhabitant of the gut of warm-blooded animals and is used frequently as an indicator bacterium of fecal contamination in the food industry. *E. coli* is a non-spore-forming, Gram-negative rod, usually motile by peritrichous flagella that is a member of the *Enterobacteriaceae* [4]. Many monitoring programs include *E. coli* because they are established markers of fecal contamination, ubiquitous in food-producing animals, easy to cultivate, and readily acquire resistance mechanisms to combat agents with activity against Gram-negative organisms [5]. They are also known reservoirs of ARGs that can be transferred horizontally to and from other closely related bacteria [6]. *E. coli* is considered a good indicator of the selective pressure imposed by antimicrobial use in food animals and has been hypothesized to be a potential predictor of emerging resistance in pathogenic bacteria that cannot be recovered from meat or animal samples Furthermore, annual trends indicate a possible correlation between *Salmonella* spp. and *E. coli* resistance [7]. The reason for using *E. coli* as an indicator is that it appears only at low background levels in the environment but possesses high survival rates [8].

Microorganisms from animal, environmental, and human sources normally contaminate raw foods [9]. The initial number of living microorganisms, including pathogens, will be substantially reduced when properly processed. However, the prevalence of pathogenic microorganisms and deterioration in ready-to-eat (RTE) foods can substantially increase through post-processing handling activities, the duration of exposure at points of sale and storage conditions [10].

At slaughter, resistant strains from the gut readily soil poultry carcasses, and as a result, poultry meat is often contaminated with resistant *E. coli*, likewise eggs become contaminated during laying [11,12,13,14,15,16,17,18]. Hence, resistant fecal *E. coli* from poultry can infect humans both directly and via food. These resistant bacteria may colonize the human intestinal tract and may also transfer resistance genes to human endogenous flora [19]. In the case of eggs, microbial contamination has a major outcome for the poultry industry and contaminated eggs are a serious public health problem worldwide. The importance of these diseases in humans can range from mild symptoms to life-threatening situations [20]. Egg and its products are an important component source of necessary nutrients. Eggs can act as a vector in the transmission of food poisoning microorganisms. Many investigations have already reported contamination of eggs with *Salmonella* spp., *Listeria monocytogenes* [21], *Campylobacter* spp. [22], and *E. coli* [23], and if the appropriate treatment does not occur, these pathogens can reach consumers’ homes and become a food safety problem.

Over the past 50 years, the use of antibiotics combined with strict biosecurity and hygiene measures has helped the poultry industry grow, preventing the negative impacts of many avian diseases caused by previously referred microorganisms [24]. The use of antibiotics to control gastrointestinal infections can lead to a change in the intestinal microbiota of hens, which can influence their immunity and health [25].

Scientific evidence suggests that the use of antimicrobials in animal production may promote bacterial resistance in treated animals [26]. Bacterial resistance of *E. coli* to antibiotics has been the subject of several studies in recent years [4,27]. Bacterial resistance to animal antibiotics is a public health problem. Antibiotic abuse and associated selection pressure led to decreased therapeutic efficacy and created populations of antibiotic-resistant microorganisms. Antibiotic resistance can spread over time, despite the suspension of antibiotic use. Several studies have suggested that antimicrobial resistance (AMR) bacteria and their AMR determinants can be transmitted from food animals to humans by direct contact and/or through animal products [28,29].

The use of antibiotics for growth promotion purposes is prohibited in the European Union. In intensive production systems, animals are exposed to a high risk of infection, as they live under stressful conditions and are driven to increase productivity. In these systems, the frequent application of antibiotics are perfect circumstances for bacterial strains to develop and resist antibiotics [30,31,32]. In Portugal, antibiotics used for application in animals, authorized for the treatment of infections, are oxytetracycline (OCT), amoxicillin (AMX), tylosin (TYL), colistin (CL), doxycycline (DOX), ampicillin (AMP), tiamulin (TIA), sulfadiazine (SFD), and enrofloxacin (ENR) [33].

*E. coli* strains have been classified based on genetic substructures associated with different phylogenies in different phylogroups that present different phenotypic and genotypic characteristics [34]. The PCR-based assay developed by Clermont et al. [35] is intended for the classification of *E. coli* strains into the major phylogroups A, B1, B2, and D; however, this method could only validate 80–85% of all *E. coli* phylogroups and it is sometimes necessary to use more alternatives [36,37]. A modification was made to the triplex method by adding one gene, resulting in a quadruple PCR [38]. Five strains or clades (I–V) were also found in *E. coli* strains, of which clade I is currently included in the phylogenetic grouping, making eight groups: A, B1, B2, C, D, E, F, and clade I [39]. Studies have shown that strains associated with virulent extraintestinal infection generally belong to phylogroups B2, D, or E and that commensal isolates of *E. coli* are generally affiliated with groups A and B1 [40,41].

Although this microorganism is considered a commensal, there are strains that can cause diarrheal diseases [42]. The virulence attributes have been used to differentiate pathogenic strains of *E. coli* and divided into diarrheal pathogens causing diarrhea (DEC) and extraintestinal *E. coli* (ExPEC) [43] based on the site of infection. There are six classic pathotypes of DEC: enteropathogenic (EPEC), shiga toxin–producing (STEC), enteroaggregative (EAEC), enterotoxigenic (ETEC), enteroinvasive (EIEC), and diffusely adherent *E. coli* (DAEC) [44]. Two additional *E. coli* pathotypes, belonging to ExPEC, are responsible for extraintestinal infections: uropathogenic *E. coli* (UPEC) causing urinary tract infections and neonatal meningitis associated *E. coli* (NMEC) [45,46]. Avian pathogenic *E. coli* (APEC) are a member of DEC, closely related to EPEC, which are frequently assigned to specific phylogenetic groups along with human UPEC and NMEC that cause disease outside the intestine [47,48]. During the last decades, the emergence of AMR bacteria has been enormously announced worldwide. In relation to an extensive use of β-lactam antibiotics in both clinical and nonclinical settings, a great diversity of β-lactamase types has consequently emerged [49]. In this context, ESBL constitute a mechanism of resistance of great clinical relevance that is spreading not only in humans but also among domestic animals [50]. ESBL-producing *Enterobacteriaceae* have been recognized as highly prevalent in food-producing animals and derived food, in the Mediterranean countries [51,52,53,54,55].

This study was conducted to investigate the prevalence and characterization of β-glucoronidase positive *E. coli* in fecal samples collected from breeding and laying hens in Portugal. The isolates were characterized by their phylogroups and a search for virulence genes was performed (pathotypes). Antimicrobial resistance assays and ESBL-associated genes detection were also performed. The main objectives were to increase knowledge regarding *E. coli* carriage in hens and to raise aware of the risks that consumers may be exposed to with the spread of MDR strains, especially if good hygiene practices are not completely fulfilled, causing a public health and/or food safety problem.

## 2. Results

### 2.1. Quantification of β-Glucoronidase Positive E. coli from Hens Fecal Samples

The microbiological load of *E. coli* in the 60 fecal samples was calculated in TBX, TBX supplemented with ampicillin (TBXamp) or with enrofloxacin (TBXenro) and the average of the results obtained were 6.03 log CFU/g in TBX, 4.77 log CFU/g in TBXamp, and 3.49 log CFU/g in TBXenro. Results of *E. coli* enumeration in the three different agar and type of hens are in shown in Table 1.

From the analysis of Table 1, it is possible to verify that there are only significant differences (*p* < 0.05) between laying hens and breeding hens feces samples in the Log TBXenro parameter, noting that this is significantly higher in the breeding hens.

### 2.2. Determination of the E. coli Phylogenetic and Patotypes Groups

Seventy nine isolates were selected for further studies as shown in Table 2.

The most predominant phylogroup was B1 (75%) followed by phylogroup A (19%) (Table 3).

Phylogroup B1 was the most predominant with more than 80% of the isolates from breeding hens and 60.7% in egg laying hens, followed by phylogroup A. Only isolates from egg laying hens were identified as phylogroup D (3.6%) or as not belonging to any phylogroup (3.6%), requiring the performance of multilocus sequence typing (MLST) that was not made. Isolates belonging to phylogroup B2, C, F, or clade I were not detected.

Concerning pathotypes, none of the isolates studied was classified as DEC.

### 2.3. Susceptibility to Antimicrobials

An overview of the antimicrobial susceptibility of the 79 *E. coli* isolates from breeding and egg laying hens is given in Table 4.

AMP was the antibiotic to which the largest number of isolates showed resistance, with 81% (64 isolates) of the resistant isolates; in decreasing order of resistance of the isolates: NAL with 65.8% (52 isolates), TET with 62.0% (49 isolates), CIP with 59.5% (47 isolates), SULF with 44.3% (35 isolates), TMP with 31.6% (25 isolates), CHL with 11.4% (9 isolates) and AZM, and CAZ and GRM with 3.8% (3 isolates). As for CTX and MEM, none of the 79 isolates from hens’ feces showed resistance, making them the only two antibiotics to which the 79 isolates were 100% sensitive.

Since the 79 hens’ feces isolates were derived from two distinct functions: breeders and layers, we decided to verify the distribution of susceptibility to the different antimicrobials as a function of the two hens’ functions (Figure 1). Excluding that for Azithromycin, a greater percentage of resistant isolates is observed in breeding hens.

### 2.4. Multiresistant Isolates

The analysis of the resistance profiles of the 79 *E. coli* isolates showed that none of the isolates under study was susceptible to all the tested antimicrobial groups; 18 (22.8%) were resistant to only one group, 16 (3.6%) were resistant to two groups, and 45 (57%) were MDR (resistant to three or more groups of antimicrobials).

The largest number of isolates (19 isolates) exhibited resistance to five different categories of antibiotics; moreover, tree isolates exhibited resistance to six groups. The PEN, FLU, TET, MA, and SULF profile was the most frequent with 42.2% (n  =  19). Table 5 summarizes the multiple MDR patterns exhibited by the 45 isolates.

When we looked at the results of *E. coli* MDR isolates, we observed that 36 isolates (80%) belong to breeding hens and only 9 belong to laying hens.

Further statistical analysis was carried in order to evaluate the existence of significant differences between ENRO- and AMP-resistant *E. coli* isolates for laying hens and breeding hens. In fact, on average, the number of antibiotic resistance observed in *E. coli* isolates for laying hens is significantly lower (*p* < 0.05) than *E. coli* isolates for breeding hens. No significant differences (*p* > 0.05) were observed between ENRO- and AMP-resistant *E. coli* isolates number of antibiotic resistance. The MDR isolates of *E. coli* within the phylogroups identified was also analyzed, and significant differences were observed between phylogroups A and B1, verifying that phylogroup A isolates presented a significant lower number of antibiotic resistance when compared with phylogroup B1 (*p* < 0.05).

### 2.5. Detection of ESBL Resistance Genes

From the isolates that showed resistance to CAZ, we verified which resistance genes were present for ESBL. In tree isolates (rate of 3.8% in analyzed samples) genes *blaTem* and *blaCTX-M* were detected. More information about the three isolates can be found in Table 6.

## 3. Discussion

*E. coli* isolates from breeding hens were more resistant to AMP (83.3%) than to ENRO (69.2%). Likewise, results from laying hens also demonstrate that the isolates were more resistant to AMP (75.1%) than to ENRO (47%). Of the total samples, isolates were more resistant to AMP (79.3%) than ENRO (58.4%), obtaining a higher number of isolates resistant to AMP than resistant to ENRO. The use of both antimicrobials is allowed for disease treatments in animals and their spread in farms may lead to an increase in microbial resistance in the population. Seventy-nine *E. coli* strains were isolated from the fecal samples of laying hens. *E. coli* strains are now classified into eight phylogroups: A, B1, B2, C, D, E, F, and clade I [41]. Studies have shown that isolates belonging to the A and B1 phylogroups are commensals, while those that belong to the B2, D, and E groups are the extraintestinal pathogenic strains [58,59,60]. Obeng et al. [61] determined the phylogenetic groups of *E. coli* isolates from the feces of intensively farmed and free-range poultry from South Australia. They found that the predominant phylogenetic groups were phylogroup B1 with 39.4% and phylogroup A with 32.3%. In a more recent study, Hayashi et al. [62], revealed that the 70 *E. coli* isolates from hens’ samples in Japan majorly belonged to group B1 (25.7%) and group A (14.3%). Similar to our results, these studies also demonstrate the predominance of B1 and A in samples from hens. In the study by Projahn et al. [63], broiler breeder lots and the corresponding eggs were analyzed. Of the eggs tested, 0.9% (n = 560) were contaminated on the outer surface of the shell. Additional analysis showed a relationship between the species found in the eggs and those isolated from the corresponding lots of origin, which demonstrates a pseudo-vertical transfer of *Enterobacteriaceae* to the hatchery. Isolates of the four positive eggs of flock were all found to be *E. coli* of the phylogroup B1. This study demonstrates the contamination of eggshells through *E. coli* contaminated feces from egg laying hens, presenting a risk to the health of consumers. It is also possible that they constitute a possible source of contamination for the chicks, given the detection of their presence in the feces of breeding hens, thus representing a risk for chicken and egg consumers. Furthermore, isolates belonging to phylogroups D and E that have been associated with virulent extraintestinal infection were found. Adefioye et al. [64] concluded in their study that most of the human isolates from fecal samples of apparently healthy individuals belonged to phylogroup B1. They only found a few isolates belonging to B2 and D phylogroups and concluded that these isolates were mostly commensals, which as a result of antibiotic exposure and other environmental and genetic factors, may revert to being pathogenic [65].

Tenaillon et al. [34] referred that genetic diversity of *E. coli* exhibits host taxonomic and environmental components. This can be illustrated by the prevalence of the four main phylogenetic groups in various human and animal populations. In humans, group A strains are predominant (40.5%), followed by B2 strains (25.5%), while B1 and D strains (17% each) are less common. In animals, there is a predominance of B1 strains (41%), followed by A strains (22%), B2 (21%) and, to a lesser extent, D strains (16%). Our results corroborate this statement since the most predominant phylogroups found in hens were B1 and A.

The presence of antibiotic-resistant foodborne pathogens in food can lead to gastrointestinal disturbances in humans [66]. On the other hand, antibiotic-resistant pathogens can transfer the gene to other microorganisms, resulting in the spread of AMR pathogens [67,68]. There are not many studies available concerning AMR in hens’ fecal samples. Two studies used a similar number of isolates from the same source. In the study by Langata et al. [69], AMR patterns among 85 resistant hen fecal isolates in Kenya were characterized and Abassi et al. [70] characterized 83 *E. coli* fecal isolates recovered from hens, in Tunisia. NAL was used in the three studies; Portugal has the higher number of resistant isolates (65.8%) while Kenya only has 18.8%. In the case of TET, the three studies presented a higher number of isolates being resistant between 90% in Tunisia and 42% in Kenya. Resistance to CTX was found in Tunisia and the number of isolates resistant to CAZ were similar in Tunisia and Portugal, and were not tested in Kenya. CIP was tested in Portugal and Kenya with 60% of Portuguese isolates being resistant and only 1.2% in Kenya. The differences found can be related to different use of antibiotics in agriculture and chicken or egg production.

In Portugal, the Directorate General for Food and Veterinary (DGAV) [33] controls the use of antibiotics and reports those authorized for the treatment of infections. Our results demonstrate antimicrobial-resistant isolates in hens’ feces that are not present in these reports, that is, that are not permitted for animal use.

When analyzing the results of antimicrobial susceptibility differentiating the types of hens between breeding and egg laying hens, we found that isolates from the feces of breeding hens showed a higher percentage of resistance to a greater number of antibiotics. Excluding that for Azithromycin, a greater percentage of resistant isolates was observed in breeding hens. This can be related to prophylactic use of antibiotics in poultry production.

Based on these results, it is not possible to make any correlation between the resistances and phylogroups.

Knowledge about MDR load and resistance patterns in isolates extracted from food-producing animals is imperative to design targeted interventions to limit antibiotic use. The use of commensal intestinal *E. coli* as a marker for the presence of resistance in bacterial flora is a critical component of MDR surveillance programs in food producing and wild animals [71]. In Liu et al. [72], fecal samples were obtained from six broiler fattening farms in China. They describe that the MDR of *E. coli* isolates was 91%. According to Koju et al. [73], hen caecum samples were collected from slaughterhouses/stores in Nepal and it was found that 71% showed resistance to at least three categories of antimicrobials. Comparing the MDR value obtained in our study (57%) with the values from those reports, we found that the number of the MDR of *E. coli* isolates from hen samples in Portugal is not as high as in other countries, despite this, it is still a worrying reality. The monitoring and treatment of drug-resistant bacteria in the poultry industry will be a long and difficult task, and one which will require a collaborative effort and should include aspects of chick breeding, the breeding environment, and feed additives.

The most common pattern of antibiotic resistance is the one that conjugates penicillins, fluoroquinolones, tetracyclines, macrolides, and sulfonamides. When we compare these data with those published by DGAV, we find that three of the categories (PEN, TET, and MA) present in the most common pattern presented by our isolates, correspond to the three classes of antibiotics most commercialized in Portugal.

Isolates from breeding hens showed higher AMR and more MDR isolates when compared to isolates from laying hens. Once again, these results demonstrate the relationship between the prophylactic use of antibiotics and poultry production.

According to the European Food Safety Authority (EFSA), the prevalence of presumptive ESBL *E. coli* producers in the different animal species and their products varies within the EU countries [74]. In Denmark, isolates of CTX-M producing *E. coli* from healthy egg laying hens (not exposed to antimicrobial agents) were found in stool samples collected from the ground and cloacal swabs, so there is a possibility to find ESBL–*E. coli* between eggshells, which can be contaminated by contact with feces [75]. In the present study, the *blaTEM* and *blaCTX-M* genes were detected in three *E. coli* isolates (5%) and could be considered as potential ESBL producers. Both the *blaTEM* and *blaCTX-M* genes have been detected among food producing meat at retail and broiler products in recent studies in Portugal [76,77]. Machado et al. [76] identified that 60% of uncooked hen carcasses (n = 20) and 10% of feces from healthy hens (n = 20) were positive for *blaTEM* and *blaCTX-M* genes. Clemente et al. [77] examined the existence of ESBL in meat collected at retail stores in Portugal and found a prevalence of 30.3% in poultry meat, while this was 11.8% and 10.5% for beef and pork, respectively. The results of these studies demonstrate the importance of monitoring the ESBL genes in *E. coli* isolates and how their study is essential for food chain safety and human health. In fact, and interesting to point out, is that the three isolates harboring ESBL genes were from three different farms, from distinct types of hens (breeding and laying) and from two different phylogroups (B1 and E) and all showing MDR. Mahmud et al. [78] have already described a high prevalence of MDR in ESBL-producing *E. coli* and 71% of ESBL-producing *E. coli* isolates were MDR. The fact that isolates came from different places indicates the potential spread of microorganisms highly resistant to antimicrobials that can reach consumers’ homes, thus changing their environmental microbiota, and if food becomes contaminated, it could accumulate and proliferate in the intestine, where genetic transfer can occur.

## 4. Conclusions

This research work complements and updates previous studies carried out in Portugal on MDR *E. coli* from food products of animal origin.

In the present study, we found *E. coli* isolates belonging mostly to phylogroup B1 and A, which are reported as commensals, but also to phylogroup E, classified as extraintestinal pathogenic strains. None of the isolates studied was classified as DEC; however, as the isolation agar used was TBX only β-glucoronidase positive *E. coli* were selected. In total, 81% of isolates tested were resistant to at least one antimicrobial and a greater percentage of resistant isolates was observed in breeding hens; 57% of total isolates were MDR and again, a greater percentage of MDR isolates was observed in breeding hens. Three resistant isolates were considered as potential ESBL producers and once again, two out of three were from breeding hens.

The work shows that *E. coli* can exhibit multiple resistance to various antimicrobials, posing a major risk to food safety and public health.

This study also reinforces previous reports that ESBL-producing *E. coli* has become one of the leading indicators for estimating MDR burden in animals and other sectors from a unique health perspective.

## 5. Materials and Methods

### 5.1. Sampling and Bacterial Isolation

Sixty fecal samples from breeding (n = 31) and egg laying (n = 29) hens were collected from the Centro and Lisboa and Vale do Tejo regions, between March and May 2019 from 36 farms, by Portuguese Food Authorities under the control programs for *Salmonella*. Laboratory procedures for the isolation and identification of *E. coli* followed the protocols defined by ISO 16649-2:2001 [79]. Briefly, 25 g of each fecal sample was mixed with 225 mL of peptone-buffered water (BPW) (Bio-Rad, Hercules, CA, USA). Decimal dilutions were prepared up to 10^−5^ with Tryptone Salt Broth (Bio-Rad) and 1 mL of the 10^−4^ and 10^−5^ dilutions were plated in Tryptone Bile X-Glucuronide (TBX) agar (Bio-Rad), 10^−3^ to 10^−5^ dilutions were plated in TBX agar supplemented with 100 μg/mL of ampicillin (TBXamp) (Sigma-Aldrich, St. Louis, MO, USA) and 10^−2^ to 10^−5^ dilutions were plated in TBX agar supplemented with 10 μg/mL of enrofloxacin (TBXenro) (Sigma-Aldrich). Ampicillin was chosen as a selective agent because it has a broad spectrum of action, particularly against Gram-negative bacteria, and is rapidly absorbed and eliminated in poultry. Enrofloxacin is applied in birds for the treatment of infections of the digestive and respiratory tract, belongs to the group of fluoroquinolones, is bactericidal, and is used against Gram-negative bacteria. Plates were incubated at 44 °C for 18 to 24 h. After the incubation period, all plates were analyzed, and the number of colonies was counted for each dilution applied. A colony from TBXamp and from TBXenro was then sub-cultured onto a slant tube containing Heart Infusion Agar (HIA) (Biogerm, Moreira, Portugal), placed again at 35 °C for 18 ± 2 h. Subsequently, these tubes were stored at 4 °C agar.

Confirmation of characteristic colonies was carried out by lactose fermentation and indol production: from the HIA a tube containing phenol red lactose broth (Sigma-Aldrich, Missouri, EUA) and a tube containing tryptophan broth (Biogerm) were inoculated and incubated at 35–37 °C for 24 h. The positive test consists of a color change from red to yellow, indicating a pH change to acidic in the lactose broth and the formation of a ring after the addition of Kovacs reagent (Sigma-Aldrich) in tryptophan broth.

### 5.2. Phylogenetic Grouping and Determination of E. coli Pathotypes

The phylogenetic group (A, B1, B2 D, E, F, and clade I) of *E. coli* isolates was determined by a specific multiplex PCR designed by Clermont et al. [38]. DNA extraction was performed using 1000 μL of presumptive *E. coli* grown overnight at 37 °C on Tryptic Soy Broth (TSB) tubes (Bio-Rad) were centrifuged for 10 min at 13,000 rpm (Bio-Rad: Model 16 Microcentrifuge), the supernatant was discarded, and the pellet was resuspended in 1000 μL of DNase-free ultra-pure water and vortexed for 2 min using BR-2000 vortexer (Bio-Rad) and the cells were lysed by boiling for 15 min using a digital heat block (VWR, Pensilvânia, EUA). The cell debris were removed by centrifugation at 10,000 rpm for 5 min using a MiniSpin microcentrifuge (ThermoFisher Scientific, Waltham, MA, USA). The supernatant was used as a template in the PCR assay in a final volume of 20 µL, containing PCR master mix multiplex PCR NZYTaq 2x Green Master mix (NZYTech, Lisbon, Portugal), and the primer set arpA (2 μM), chuA (1 μM), yjaA (1 μM), and TspE4.C2 (1 μM) (Eurofins Genomics, Porto, Portugal) as described in Table 7. For the PCR reaction, we considered the number of samples to be validated, positive controls, and negative control. *E. coli* O111 (arpA^+^; TspE4.C2^+^), *E. coli* O157:H7 (arpA^+^; chuA^+^), and *E. coli* K12 (arpA^+^; yjaA^+^) were used as positive controls. As a negative control, in place of the template, DNase-free water was added in the same amount. The PCR conditions were as follows: 95 °C, 3 min; 39 cycles of 95 °C, 30 s; 58 °C, 30 s; 72 °C, 30 s; and 72 °C, 5 min. The results were visualized using UV light (Syngene^®^ Cambridge, UK).

According to the presence or absence of *arpA, chuA*, *yjaA*, and *TspE4.C2* genes, a phylogroup was assigned to each isolate, as previously described by Clermont et al. [38] Table 8.

For the distinction of phylogroups, which could be group A or C, E or D, or E, or clade I, two more PCR were performed.

For the distinction of phylogroups A or C, 1 µL of template was used for PCR in a final volume of 20 µL, containing MgCl_2_ (1.5 mM), Taq buffer (1x), and trpA (0.25 μM) (Eurofins Genomics) as described in Table 9, Taq (1U), (NZYTech) dNTPs (0.25 μM) (NZYTech) and DNase-free water. DNA extraction was performed as previously described. For the PCR reation, the number of samples to be validated for this phylogenetic group was considered, a positive and a negative control. *E. coli* FV19459 (trpA^+^) was used as a positive control. As a negative control, in place of the template, DNase-free water was added in the same amount. The PCR conditions were as follows: 95 °C, 5 min; 30 cycles of 95 °C, 30 s; 59 °C, 30 s; 72 °C, 6 s; and 72 °C, 5 min. The results were visualized using UV light (Syngene^®^).

To distinguish between E or D, or E or I clade, 2.4 µL of template was used for PCR in a final volume of 20 µL, containing PCR master mix multiplex PCR NZTYtaq 2x Green Master Mix (NZYTech), arpA (2 μM) (Eurofins Genomics) as described in Table 10 and DNase-free water were used. DNA extraction was performed as previously described. For the PCR reaction, were considered the number of samples to be validated, a positive and a negative control. *E. coli* O157:H7 (arpA^+^) was used as a positive control. As a negative control, in place of the template, DNase-free water was added in the same amount. The PCR conditions were as follows: 95 °C 5 min; 30 cycles of 95 °C, 30 s; 57 °C, 30 s; 72 °C, 9 s; and 72 °C, 5 min. The results were visualized using UV light (Syngene^®^).

For the elaboration and determination of the pathotypes (ETEC, EIEC, EAEC, EPEC, EHEC/STEC) the primers (Eurofins Genomics) designated by Schmidt et al. [80], Aranda et al. [81] and ISO/TS 13136 [82] were used. DNA extraction was performed as previously described. As positive controls *E. coli* IH2859f (eae+; bfp+), *E. coli* LMV_E_37 (eae+; bfp+), *E. coli* LMV_E_38 (est+), *E. coli* LMV_E_39 (12 et+), *E. coli* LMV_E_40 (ipaH+), *E. coli* LMV_E_41 (aggr+; cvd432+), *E. coli* O157.34 (eae+; stx1+; stx2+), *E. coli* O157.156 (eae+; stx1+; stx2+) and *E. coli* O157.157 (eae+; stx1+; stx2+) were used. As a negative control, in place of the template, DNase-free water was added in the same amount. All gene amplifications were obtained by multiplex PCR. For all assays, the master mix used was the multiplex PCR NZTYtaq 2x Green Master mix (NZYTech). For all assays, the total reaction volume was 20 μL. The list of primers used to determine each pathotype, their concentrations, amplification conditions, and volume per reaction are described in Table 11 and Table 12. The results were visualized using UV light (Syngene^®^).

### 5.3. Antimicrobial Susceptibility of E. coli Isolates

Antimicrobial susceptibility was tested by disk-diffusion method on Mueller–Hinton agar (Mha) (Bio-Rad). Briefly, a suspension of *E. coli* in saline solution (NaCl 0.85%) with a 0.5 McFarland turbidity was prepared. A sterile swab was dipped into this suspension, swirled well against the walls of the tube to remove excess solution, and used to inoculated by streaking Mha (Bio-Rad) plate. The disks of the antimicrobials were placed onto the surface and lightly pressed. Plates were incubated at 37 °C for 20 ± 4 h. At the end of incubation, the inhibition zone diameters were measured with a ruler. The antibiotics tested were as follows: AMP (10 μg) (Bio-Rad), CTX (5 μg) (Bio-Rad), CAZ (10 μg) (Bio-Rad), MEM (10 μg) (Oxoid, Basingstoke, UK), NAL (30 μg) (Oxoid), CIP (5 μg) (Oxoid), GEN (10 μg) (Bio-Rad), AZM (15 μg) (Oxoid), TE (30 μg) (Oxoid), TMP (5 μg) (Bio-Rad), CHL (30 μg) (Bio-Rad), and SULF (300 μg) (Oxoid). The breakpoints for AMP, CTX, CAZ, MEM, NAL, CIP, GEN, AZM, TE, TMP and CHL fitted the susceptibility profile according to EUCAST [56] parameters. SULF fitted the susceptibility profile according to CLSI [57] since EUCAST does not show breakpoint values. Reference strains *E. coli* ATCC 25922, *S. aureus* ATCC 25923, and *P. aeruginosa* ATCC 27853 were adopted as control strains. The results of the susceptibility profile are presented in break intervals defined in two categories (S—susceptible and R—resistant).

### 5.4. Multiresistant Isolates of E. coli Assessment

According to Magiorakos et al. [83], to characterize MDR bacteria it is necessary that the bacteria show an acquired non-susceptibility to at least one agent in three or more antimicrobial groups. The antimicrobial groups considered were penicilins, cephalosporins, carbapenems, fluoroquinolones, aminoglycosides, macrolides, tetracyclines, miscellaneous agents, and sulfonamides. Isolates that showed resistance to at least three different antimicrobial groups were classified as MDR.

### 5.5. Detection of Extended-Spectrum β-Lactamase Resistance Genes

The isolates that exhibited resistance to CAZ were analyzed by multiplex PCR using primers described in Table 13, targeting genes blaCTX-M, blaTEM, and blaSHV [84,85,86]. The DNA template was obtained as described in 4.2; 2 µL of the DNA suspension was used for PCR in a final volume of 20 µL, containing 1.5 mM MgCl_2_, 1x Taq buffer, 0.25 μM primers, 1.25 U Taq (NZYTech), 0.25 mM dNTPs (NZYTech), and DNase-free water. The PCR conditions were as follows, according to Oliveira et al. [87]: 95 °C, 5 min; 30 cycles of 94 °C, 30 s; 60 °C, 30 s; 72 °C, 30 s; and 72 °C, 5 min. A 2% agarose gel (NZYTech) in 1x TAE (NZYTech) was prepared containing 5 µL/100 mL of GreenSafe Premium (NZYTech). The gel was loaded with the reaction products: 4 μL of 6x NZYDNA loading dye (NZYTech) was added to the amplicons that did not contain a loading dye. *E. coli* R02 (blaCTX-M^+^, blaTEM^+^); *E. coli* H1015 (blaSHV-12^+^); *E. coli* H1043 (blaCTX-M-I^+^); *E. coli* H1046 (blaCTX-M-II^+^, blaTEM^+^), and *E. coli* H995 (blaCTX-M-IX^+^, blaTEM^+^) were used as positive controls. As a negative control, in place of the template, DNase-free water was added in the same amount. Each corresponding well was loaded with 8 μL of the reaction obtained after amplification and with 5 μL V-marker for ESBL (NZYTech) and finally the gel ran in 1x TAE (NZYTech) at 90 volts ± 10 volts for 1 h and visualized using UV light (Syngene^®^ GeneFlash system).

### 5.6. Data Analysis

Descriptive statistics (mean, standard deviation (SD)) were generated for all of the variables in the dataset. Analysis of variance was performed on microbiological load of *E. coli* in TBX, TBX supplemented with ampicillin (TBXamp), or with enrofloxacin (TBXenro), for breeding hens and egg laying hens’ feces’ samples data, using the IBM SPSS Statistics 23.0 for Windows [88]. The analysis was carried out using a *t*-test of independent samples with a variable grouping of sample sources (breeding hens and egg laying hens). All statements of significance were based on testing at the *p* < 0.05 level.

## Figures and Tables

**Figure 1 antibiotics-12-00020-f001:**
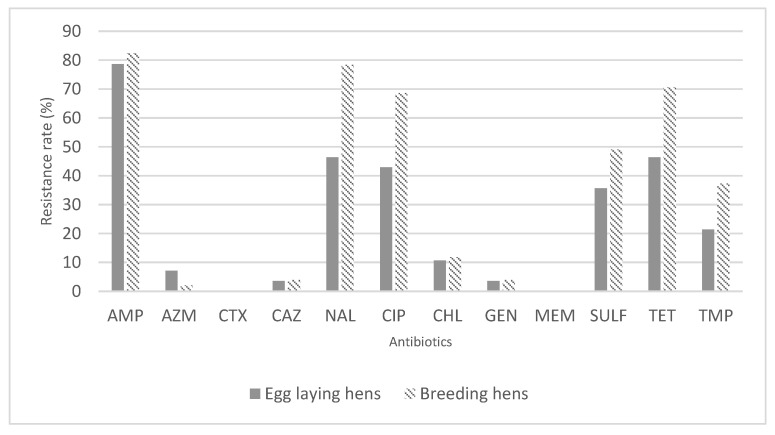
Percentage of resistance to the antimicrobials under study, for feces isolates of *E. coli* from egg laying hens or breeding hens.

**Table 1 antibiotics-12-00020-t001:** Microbiological load of *E. coli* in TBX, TBX supplemented with ampicillin (TBXamp) or with enrofloxacin (TBXenro), for breeding hens and egg laying feces samples (mean ± standard deviation).

Sample	Samples Sources	Microbiological Load of *E. coli* (log CFU/g)
Log_TBX	Egg laying hens	6.02 a ± 1.43
	Breeding hens	6.03 a ± 0.98
Log_TBXamp	Egg laying hens	4.46 a ± 1.65
	Breeding hens	5.06 a ± 1.63
Log_TBXenro	Egg laying hens	2.81 a ± 1.12
	Breeding hens	4.12 b ± 1.33

a, b—Values with different letters within the same column indicate significant differences (*p* ≤ 0.05) between samples sources.

**Table 2 antibiotics-12-00020-t002:** Number of samples of laying and breeding hens analyzed, and number of isolates of *E. coli* resistant to ampicillin or enrofloxacin selected for further characterization.

Samples Source	No. of Samples	No. of Isolates	No. of Isolates AMP (R)	No. of Isolates ENR (R)
Breeding hens	31	51	21 (41.2%)	30 (58.8%)
Egg laying hens	29	28	16 (57.1%)	12 (42.9%)
Total	60	79	37 (46.8%)	42 (53.2%)

(R) Resistant.

**Table 3 antibiotics-12-00020-t003:** Percentages of phylogroup detected in isolates from hens feces.

	Phylogroup				
	A	B1	D	E	Unknown
No. of isolates of breeding hens (%)	8 (15.6%)	42 (82.4%)	0 (0%)	1 (2%)	0 (0%)
No. of isolates of egg laying hens (%)	7 (25%)	17 (60.7%)	1 (3.6%)	2 (7.1%)	1 (3.6%)
Total no. of isolates (%)	15 (19%)	59 (74.6%)	1 (1.3%)	3 (3.8%)	1 (1.3%)

**Table 4 antibiotics-12-00020-t004:** Susceptibility to antibiotics of *E. coli* isolated from feces of egg laying and breeding hens according to EUCAST [56] and CLSI [57] parameters.

Antibiotic	No. of Isolates (%S)	No. of Isolates (%R)
AMP	15 (19.0)	64 (81.0)
AZM	76 (96.2)	3 (3.8)
CTX	79 (100.0)	0 (0.0)
CAZ	76 (96.2)	3 (3.8)
NAL	27 (34.2)	52 (65.8)
CIP	32 (40.5)	47 (59.5)
CHL	70 (88.6)	9 (11.4)
GEN	76 (96.2)	3 (3.8)
MEM	79 (100.0)	0 (0.0)
SULF	44 (55.7)	35 (44.3)
TET	20 (38.0)	49 (62.0)
TMP	54 (68.4)	25 (31.6)

AMP—ampicillin; AZM—Azithromycin; CTX—Cefotaxime; CAZ—Ceftazidime; NAL—Nalidixic acid; CIP—Ciprofloxacin; CHL—Chloramphenicol; GEN—Gentamicin; MEM—Meropenem; SULF—Sulfonamides; TET—Tetracycline; TMP—Trimethoprim; (R) resistant; (S) susceptible.

**Table 5 antibiotics-12-00020-t005:** Distribution of multi resistant isolates of *E. coli* and number of categories for which they exhibited resistance.

No. of Antimicrobials Classes Per Group	MDR Pattern	No. of Isolates of Breeding Hens	No. of Isolates of Egg Laying Hens	Total No. of Isolates (%)
6	PEN + CEP + FQs + TETs + MA + SULFs	1 0 0	0 1 1	3 (6.7%)
	PEN + FQs + M + TETs + MA + SULFs			
	PEN + FQs + AMG + TETs + MA + SULFs			
5	PEN + FQs + TETs + MA + SULFs PEN + AMG + TETs + MA + SULFs FQs + MGS + TETs + MA + SULFs	14 1 1	2 0 1	19 (42.2%)
4	PEN + CEP + FQs + TETs	0	1	9 (20%)
	PEN + FQs + TETs + SULFs	5	0	
	PEN + FQs + TETs + MA	1	1	
	PEN + TETs + MA + SULFs	1	0	
3	PEN + CEP + TETs PEN + FQs + SULFs PEN + FQs + TETs PEN + TETs + MA PEN + TETs + SULFs PEN + MA + SULFs FQs + TETs + SULFs	1 6 1 0 3 0 1	0 0 0 1 0 1 0	14 (31.1%)
Total		36	9	45 (100%)

PEN—penicillins; CEP—cephalosporins; C—carbapenems; FQs—fluoroquinolones; AMG—aminoglycosides; M—macrolides; TETs—tetracyclines; MA—miscellaneous agents; SULFs—sulfonamide.

**Table 6 antibiotics-12-00020-t006:** Characteristics of isolates positive for resistant ESBL genes.

Isolate	Type	Phylogroup	AMR	MDR
5 AMP	Breeding	E	AMP + CAZ + NAL + CIP + SULF + TET + TMP	6 classes
15 AMP	Breeding	B1	AMP + CAZ + TET	4 classes
29 ENRO	Egg laying	B1	AMP + CZD + NAL + CIP + TET	3 classes

**Table 7 antibiotics-12-00020-t007:** List of primers used for the determination of phylogenetic groups.

Primer	Target	Sequence (5′-3′)	PCR Product. (bp)	Ref.
arpA fwd	*arpA*	AACGCTATTCGCCAGCTTGC	400	[38]
arpA rev		TCTCCCCATACCGTACGCTA		
chuaA fwd	*chuaA*	ATGGTACCGGACGAACCAAC	288	[38]
chuaA rev		TGCCGCCAGTACCAAAGAC		
yjaA fwd	*yjaA*	CAAACGTGAAGTGTCAGGAG	211	[38]
yjaA rev		AATGCGTTCCTCAACCTGTG		
TspE4.C2	*TspE4.C2*	CACTATTCGTAAGGTCATCC	152	[38]
TspE4.C2 rev		AGTTTATCGCTGCGGGTCGC		

**Table 8 antibiotics-12-00020-t008:** Assignment of phylogroups of *E. coli* isolates based on the presence of genes *arpA*, *chuA*, *yjaA*, and *TspE4.C2*.

Phylogroup	Target Gene			
	*arpA*	*chuA*	*yjaA*	*TspE4.C2*
A	+	-	-	-
A or C	+	-	+	-
B1	+	-	-	+
B2	-	+	+	-
B2	-	+	-	+
B2	-	+	+	+
E or D	+	+	-	-
E or D	+	+	-	+
E or Clade I	+	+	+	-
F	-	+	-	-
(a)	+	-	+	+

It is necessary to perform MLST to identify the phylogenetic group.

**Table 9 antibiotics-12-00020-t009:** List of primers used for the distinction of phylogroups A or C.

Primer	Target	Sequence (5′-3′)	PCR Product. (bp)	Ref.
trpAgpC.1 fwd	*trpA*	AGTTTTATGCCCAGTGCGAG	219	[38]
trpAgpC.1 rev		TCTGCGCCGGTCACGCCC		

**Table 10 antibiotics-12-00020-t010:** List of primers used for the distinction of phylogroups E or D, or E or I clade.

Primer	Target	Sequence (5′-3′)	PCR Product. (bp)	Ref.
ArpAgpE fwd	*arpA*	GATTCCATCTTGTCAAAATATGCC	301	[38]
ArpAgpE rev		GAAAAGAAAAAGAATTCCCAAGAG		

**Table 11 antibiotics-12-00020-t011:** List of primers used for the determination of pathotypes.

Pathotypes	Primer	Target	Sequence (5′-3′)	PCR Product (bp)	Ref.
ETEC	est (ST) fwd	*est (ST)* *elt (LT)*	ATTTTTMTTTCTGTATTRTCTT	190 450	[81]
	est (ST) rev		CACCCGGTACARGCAGGATT		
	elt (LT) fwd elt (LT) rev		GGCGACAGATTATACCGTGC CGGTCTCTATATTCCCTGTT		
EIEC	ipaH fwd ipaH rev	*ipaH*	GTTCCTTGACCGCCTTTCCGATACCGTC GCCGGTCAGCCACCCTCTGAGAGTAC	600	[81]
EAEC	aggr fwd aggr rev cvd432 fwd cvd432 rev	*aggr* *cvd432*	GTATACACAAAAGAAGGAAGC ACAGAATCGTCAGCATCAGC CTGGCGAAAGACTGTATCAT CAATGTATAGAAATCCGCTGTT	254 630	[80]
EPEC	bfpA fwd bfpA rev eae fwd eae rev	*bfpA* *eae*	AATGGTGCTTGCGCTTGCTGC GCCGCTTTATCCAACCTGGTA GACCCGGCACAAGCATAAGC CCACCTGCAGCAACAAGAGG	326 384	[81]
STEC	stx1 fwd stx1 rev stx2 fwd stx2 rev eae fwd eae rev	*stx1* *stx2* *eae*	ATAAATCGCCATTCGTTGACTAC AGAACGCCCACTGAGATCATC CGCACTGTCTGAAACTGCTCC TCGCCAGTTATCTGACATTCTG GACCCGGCACAAGCATAAGC CCACCTGCAGCAACAAGAGG	180 255 384	[82]

**Table 12 antibiotics-12-00020-t012:** List of concentrations of each primer, amplification conditions.

Primer	Primer Concentration	Amplification Conditions
est (ST) fwd	0.5 μM 0.5 μM 0.5 μM 0.5 μM	95 °C, 5 min 35 cycles of 95 °C, 30 s; 55 °C, 60 s; 72 °C, 14 s 72 °C, 5 min
est (ST) rev		
elt (LT) fwd elt (LT) rev		
ipaH fwd ipaH rev	0.2 μM 0.2 μM	95 °C, 5 min 30 cycles of 95 °C, 30 s; 60 °C, 1 min; 72 °C, 18 s 72 °C, 5 min
aggr fwd aggr rev	0.2 Mm 0.2 Mm	95 °C, 5 min 30 cycles of 95 °C, 30 s; 60 °C, 1 min; 72 °C, 8 s 72 °C, 5 min
cvd432 fwd cvd432 rev	0.2 μM 0.2 μM	95 °C, 5 min 10 cycles of 95 °C, 30 s; 55 °C, 1 min; 72 °C, 19 s 20 cycles of 95 °C, 30 s; 60 °C, 1 min; 72 °C, 19 s 72 °C, 5 min
bfpA fwd bfpA rev	0.2 μM 0.2 μM	95 °C, 5 min 10 cycles of 95 °C, 30 s; 55 °C, 1 min; 72 °C, 10 s 20 cycles of 95 °C, 30 s; 60 °C, 1 min; 72 °C, 10 s 72 °C, 5 min
stx1 fwd stx1 rev stx2 fwd stx2 rev eae fwd eae rev	0.8 μM 0.8 μM 2.4 μM 2.4 μM 0.8 μM 0.8 μM	95 °C, 5 min 9 cycles of 95 °C, 60 s; 65 °C, 2 min; 72 °C, 90 s 95 °C, 60 s; 64 °C, 2 min; 72 °C, 90 s; 95 °C, 60 s; 63 °C, 2 min; 72 °C, 90 s 95 °C, 62 s; 64 °C, 2 min; 72 °C, 90 s; 95 °C, 60 s; 61 °C, 2 min; 72 °C, 90 s 10 cycles of 95 °C, 60 s; 60 °C, 2 min; 72 °C, 90 s 9 cycles of 95 °C, 60 s; 60 °C, 2 min; 72 °C, 150 s 72 °C, 5 min

**Table 13 antibiotics-12-00020-t013:** List of primers used for detection of extended-spectrum β-lactamase resistance genes.

Primer	Target	Sequence (5′-3′)	PCR Product. (bp)	Ref.
blaTEM fwd	*blaTEM*	CATTTCCGTCGCCCTTATTC	800	[84]
blaTEM rev		CGTTCATCCATAGTTGCCTGAC		
blaSHV fwd	*blaSHV*	AGCCGCTTGAGCAAATTAAAC	713	[85]
blaSHV rev		ATCCCGCAGATAAATCACCAC		
blaCTX-M fwd	*blaCTX-M*	ATGTGCAGYACCGTAARGTKATGC	593	[86]
blaCTX-M rev		TGGGTRAARTARGTSACCAGAAYCAGCGG		

## Data Availability

The data presented in this study are available on request from the corresponding author.

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
