# Peer review of "Evaluation of Antimicrobial Resistance of Different Phylogroups of Escherichia coli Isolates from Feces of Breeding and Laying Hens"

_antibiotics, 2022, doi:10.3390/antibiotics12010020_

Round 1
Reviewer 1 Report
The article is about the evaluation of antimicrobial resistance of different phylogroups of Escherichia coli from feces of breeding and laying hens. The article is interesting, however, it presents several errors in writing and spelling, which makes it difficult to read and understand. As only low numbers of E. coli were analyzed, associations between phylogroups, AMR and genotypes cannot be made. In my opinion, the overall output has several weak points that prevent publication of the paper. In details, the following consideration should be taken into account to eventually reconsider a completely new submission.
- Line 3: Title: Put the word “isolates” after Escherichia coli.
- Line 19: Change “E.coli” to “Echerichia coli”
- Line 47: Remove “family”
- Line 55: Put spp. In front of the “Salmonella”.
- The abuse and misuse of antibiotics in animals contributes to the rise of antibiotic resistance. The authors should briefly state which antibiotics are used in the treatment of breeding and egg laying hen infections.
- The purpose and novelty of this study must be better explained.
Results
- Line 112: “E. coli” instead of “Escherichia coli”
- Lines 115 and 116: “CFU/ g” please delete space before “g”.
- Table 2. Please reconstruct the table. The word “phylogroup” should be placed on the top of the phylogroup columns.
- Table 3: In the second and third columns, you must first present the number of isolates and then their percentages.
- Figure 1: Y axis: “Resistance rate (%)” instead of “Percentage of resistance exhibited”.
- Table 4: Please present antimicrobial profiles of MDR isolates.
- Line 165: Please change “Escherichia coli” to “E. coli”.
- Line 171 and elsewhere: “blaTem” and “blaCTX-M” genes in italic.
Discusion and Conclusion
- The aim of the study should be stated in the introduction section.
- Please strengthen the discussion of the manuscript.
-Conclusion has been described very poorly. It should be improved.
Material and methods
- You should report all details of materials (Supplier, purity and ….)
- Section 5.1.: Please explain the sampling method.
- Line 263: Please change “Escherichia coli” to “E. coli”
- Line 266: “with” instead of “within”
- Line 267 and 268: “agar” instead of “medium”.
- Line 274-275: The incubation time for Enterobacteriaceae is 24-48 h. Why is 20 h chosen?
- Line 276: “Isolates that showed resistance were…” Explain more clearly here. Resistance to what?
- Line 288 and elsewhere: The names of the primers should be written normally and not in italics.
- Line 288: Please provide the primers used in the study with their specifications in a table
- Lines 289-291: Write “The PCR conditions were as follows : 95°C 3 min ; 39 cycles of 94°C 30 s, 58°C 30 s, 72°C 30 s, and 72°C 5 min.”
- Lines 307-313: Please provide the antimicrobial content of each disc.
- Lines 311-316: Please rewrite the sentences.
- Lines 322-328: This sentence is mentioned in the previous section (section 5.3.), so it is repetitive.
- Lines 336 and 337: Write “The PCR conditions were as follows : 95°C 5 min ; 35 cycles of 94°C 30 s, 60°C 30 s, 72°C 30 s, and 72°C 5 min.”
- Line 338: “PCR” instead of “reaction”
- Section 5.5.: For identification of ESBL resistance genes, please provide the positive and negative control used.
- Add the statistical analysis section. Describe the study factors and variables.
Author Response
Thank you for the valuable revision and comments on our paper. The updated version was reformulated according to the reviewer's suggestions. The lines number are related to the file with the track changes.
Revisor #1
- Line 3: Title: Put the word “isolates” after Escherichia coli.
Authors agreed and added the term to the title.
- Line 19: Change “E.coli” to “Echerichia coli”
R:Change of the term was made.
- Line 48: Remove “family”
R: The authors agreed and removed the term.
- Line 55: Put spp. In front of the “Salmonella”.
R: Authors agreed and added spp.
- The abuse and misuse of antibiotics in animals contributes to the rise of antibiotic resistance. The authors should briefly state which antibiotics are used in the treatment of breeding and egg laying hen infections.
R: Authors agreed and added a paragraph, Line 99: “The use of antibiotics for development is prohibited in the European Union, but in countries outside Europe it appears that it is applied. In intensive production systems, animals are exposed to a high risk of infection, as they live under stressful conditions and are driven to increase productivity. In this system, the frequent application of antibiotics acquires perfect circumstances for strains of bacteria to develop and resist antibiotics. In Portugal, antibiotics for application in chickens, authorized for the treatment of infections, are amoxicillin (AMX), enrofloxacin (ENX), tiamulin (TIA), doxycycline (DOX), oxytetracycline (OTC), neomycin (NEO) and tylosin (TYL)”.
- The purpose and novelty of this study must be better explained.
R: purpose and novelty were improved Line 142
Results
- Line 112: “E. coli” instead of “Escherichia coli”
R: Authors agreed to change the term.
- Lines 152 and 153: “CFU/ g” please delete space before “g”.
R: corrections have been made.
- Table 3. Please reconstruct the table. The word “phylogroup” should be placed on the top of the phylogroup columns.
R: Authors agreed with restructuring the table and some corrections were made
- Table 2: In the second and third columns, you must first present the number of isolates and then their percentages.
R: Authors agreed to change the way they present the results in the table.
- Figure 1: Y axis: “Resistance rate (%)” instead of “Percentage of resistance exhibited”.
R: changes have been made.
- Table 4: Please present antimicrobial profiles of MDR isolates.
R: A new table was made with antimicrobial profiles of MDR isolates.
- Line 165: Please change “Escherichia coli” to “E. coli”.
R: change has been made.
- Line 171 and elsewhere: “blaTem” and “blaCTX-M” genes in italic.
R: Authors agree with the change.
Discusion and Conclusion
- The aim of the study should be stated in the introduction section.
R: The aim of the study was rewritten and it is in the introduction section
- Please strengthen the discussion of the manuscript.
R: Discussion was improved.
-Conclusion has been described very poorly. It should be improved.
R: Conclusions were improved.
Material and methods
- You should report all details of materials (Supplier, purity and ….)
R: More details were added.
- Section 5.1.: Please explain the sampling method.
R: More details were added explaining the sampling method, however sampling was performed under the program to reduce Salmonella and was performed by food authorities.
- Line 263: Please change “Escherichia coli” to “E. coli”
R: Change has been made
- Line 266: “with” instead of “within”
R: Change has been made.
- Line 267 and 268: “agar” instead of “medium”.
R: Change has been made
- Line 274-275: The incubation time for Enterobacteriaceae is 24-48 h. Why is 20 h chosen?
R: Line 408 We didn’t perform the enumeration of Enterobacteriaceae we have performed the determination of β-glucoronidase positive E. coli following the ISO 16640-2 standard where the incubation time should be 18-24h at 44ºC. So we have corrected the methodology described.
- Line 276: “Isolates that showed resistance were…” Explain more clearly here. Resistance to what?
R: Line 409 The sentence has been changed to “A colony from TBXamp and from TBXenro was then sub-cultured onto a slant tube…”.
- Line 288 and elsewhere: The names of the primers should be written normally and not in italics.
R: Authors agreed to write normally the names of the primers.
- Line 288: Please provide the primers used in the study with their specifications in a table
R: A new table was added with the specific information. Table 5.
- Lines 289-291: Write “The PCR conditions were as follows : 95°C 3 min ; 39 cycles of 94°C 30 s, 58°C 30 s, 72°C 30 s, and 72°C 5 min.”
R: Authors agree to change the way PCR conditions are described.
- Lines 307-313: Please provide the antimicrobial content of each disc.
R: The information was added.
- Lines 311-316: Please rewrite the sentences.
R: Authors agree with rewrite the sentences.
- Lines 322-328: This sentence is mentioned in the previous section (section 5.3.), so it is repetitive.
R: Authors agree. The sentence was deleted.
- Lines 336 and 337: Write “The PCR conditions were as follows : 95°C 5 min ; 35 cycles of 94°C 30 s, 60°C 30 s, 72°C 30 s, and 72°C 5 min.”
R: Authors agree to change the way PCR conditions are described.
- Line 338: “PCR” instead of “reaction”
R: Authors agreed with the change of the term.
- Section 5.5.: For identification of ESBL resistance genes, please provide the positive and negative control used.
R: The information was added. Table 8.
- Add the statistical analysis section. Describe the study factors and variables.
R: The author added section 5.6 Data analysis
Reviewer 2 Report
Authors reported a study regarding the antimicrobial resistance of E. coli strains isolated from egg laying and breeding hens. They evaluated pathotypes, antimicrobial resistance, and resistance genes associated with ESBL.
The results obtained are interesting, but the manuscript must be improved.
English version must be revised
Abstract
Abstract must be of about 200 words maximum. Authors must be reduce the paragraph.
Introduction
The introduction is well written
Results
2.1. Quantification of Escherichia coli in samples from primary production animals: I suggest changing the title… the quantification concerns stool samples.
You write about MLST but there is no mentioned in the materials and methods. It is not clear why it is not specified that MLST was used only for strains whose phylogroup as not able to be defined.
Table 2: Phylogroup is written in the wrong column: that's the number strains. MLST entry must be deleted and replaced with ND (not determinated)
MLST results are not present: I suggest to add a table
Table 3: I suggest putting percentages in brackets and not the number of isolates
Lane 147: “(category R)” is superfluous
Figure 1: ordinate axis: it is not the percentage of resistance, but the percentage of resistant strains
Is there a difference statistically significant between the two categories? Furthermore, I would add a sentence describing what is observed: excluding that for Azithromycin a greater percentage of resistant isolates is observed in Breeding hens
2.4. Multiresistant isolates: this paragraph is not exhaustive. In addition to the percentages of isolates resistant to the antibiotic categories, it is necessary to indicate which are the most widespread associations between categories. It would give important information. How are these multi-resistors located within the two groups?
2.5. Detection of ESBL resistance genes: were the 4 strains isolated from Egg laying hens or Breeding hens? Were they MDR strains? I suggest to expand this paragraph with this information
Discussion
MDR results and which are the most common is not properly discussed. I would re-evaluate the discussion after a better presentation of the results
Material and method
Lane 288: I suggest putting concentrations of each primer next to genes, in brackets.
Lane 294: Please, indicate the base pair obtained per gene
Lane 160: delete "approximately"
Lane 295: ETEC, EIEC, EAEC, EPEC, EHEC / STEC I suggest to put it in brackets and add "were used" at the end of the sentence. Also, I suggest to add a table with primers used, fragment size (bp) and references
Lane 302: "a suspension in saline solution (NaCl 0.85%) with a 0.5 McFarland turbidity…" A suspension of E. coli? Please, specify
5.3. Antimicrobial susceptibility of Escherichia coli isolates: please specify the concentration for any antibiotic used
Lane 323: please, put only the acronyms of the antibiotics ... the complete name it is already written above
Lane 326: I wouldn't really agree with categorizing tetracyclines as streptogramins
Lane 331-332: the sentence is not clear, perhaps there are some typos left.
Lane 333: I suggest removing this sentence. It has already been said previously how the DNA was extracted.
Lane 336: I suggest to put 0.25 μM outside brackets like the concentrations written above
Author Response
Thank you for the valuable revision and comments on our paper. The updated version was reformulated according to the reviewer's suggestions. The lines number are related to the file with the track changes.
Revisor #2
Abstract
Abstract must be of about 200 words maximum. Authors must be reduce the paragraph.
R: The paragraph was reduce.
Introduction
The introduction is well written
Results
2.1. Quantification of Escherichia coli in samples from primary production animals: I suggest changing the title… the quantification concerns stool samples.
R: Authors agreed with changing the title. New title: “Quantification of E. coli from hens fecal samples”
You write about MLST but there is no mentioned in the materials and methods. It is not clear why it is not specified that MLST was used only for strains whose phylogroup as not able to be defined.
R: MSLT has never being performed, it is described when there is not possible to assign a phylogroup based in the profile obtained with PCR. The authors removed this information from the article and rewrote the sentence.
Table 2: Phylogroup is written in the wrong column: that's the number strains. MLST entry must be deleted and replaced with ND (not determined)
R: Authors agreed with restructuring the table.
MLST results are not present: I suggest to add a table
R: As mentioned before MSLT was never done. The authors removed this information from the article and rewrote the sentence.
Table 3: I suggest putting percentages in brackets and not the number of isolates
R: Authors agreed to change the way they present the results in the table.
Lane 147: “(category R)” is superfluous
R: Authors agreed.
Figure 1: ordinate axis: it is not the percentage of resistance, but the percentage of resistant strains
R: Authors agree with change.
Is there a difference statistically significant between the two categories? Furthermore, I would add a sentence describing what is observed: excluding that for Azithromycin a greater percentage of resistant isolates is observed in Breeding hens
R: Authors agreed that the information is relevant and added the information.
2.4. Multiresistant isolates: this paragraph is not exhaustive. In addition to the percentages of isolates resistant to the antibiotic categories, it is necessary to indicate which are the most widespread associations between categories. It would give important information. How are these multi-resistors located within the two groups?
R: Authors agreed. A new table (Table 4) was added to the results with the necessary information and the results were discussed.
2.5. Detection of ESBL resistance genes: were the 4 strains isolated from Egg laying hens or Breeding hens? Were they MDR strains? I suggest to expand this paragraph with this information
R: New information was added (table 5).
Discussion
MDR results and which are the most common is not properly discussed. I would re-evaluate the discussion after a better presentation of the results
R: Authors agreed. the results were better explained, and the discussion was improved.
Material and method
Lane 288: I suggest putting concentrations of each primer next to genes, in brackets.
R: Concentrations of each primer were put next to genes, in brackets as suggested.
Lane 294: Please, indicate the base pair obtained per gene
R: The information was added.
Lane 160: delete "approximately"
R: Authors agreed with the elimination.
Lane 295: ETEC, EIEC, EAEC, EPEC, EHEC / STEC I suggest to put it in brackets and add "were used" at the end of the sentence. Also, I suggest to add a table with primers used, fragment size (bp) and references
R: Authors agreed to put the designations in brackets and add "were used" at the end of the sentence. A table was added with the information required. Table 6.
Lane 302: "a suspension in saline solution (NaCl 0.85%) with a 0.5 McFarland turbidity…" A suspension of E. coli? Please, specify
R: Yes, a suspension of E. coli. The information was added.
5.3. Antimicrobial susceptibility of Escherichia coli isolates: please specify the concentration for any antibiotic used
R: All of the concentrations for any antibiotic used were added.
Lane 323: please, put only the acronyms of the antibiotics ... the complete name it is already written above
R: Authors agreed with eliminating the complete name.
Lane 326: I wouldn't really agree with categorizing tetracyclines as streptogramins
R: Neither the authors, that’s why tetracycline was categorized as tetracyclines. The only antimicrobial that was characterized as “macrolides, glycosamides and streptogramins” was azithromycin.
Lane 331-332: the sentence is not clear, perhaps there are some typos left.
R: The sentence was rewriting for better comprehension.
Lane 333: I suggest removing this sentence. It has already been said previously how the DNA was extracted.
R: Authors agreed with the removal of the sentence.
Lane 336: I suggest to put 0.25 μM outside brackets like the concentrations written above
R: Authors agreed.

Reviewer 3 Report
Evaluation of antimicrobial resistance of different phylogroups of Escherichia coli from feces of breeding and laying hens.
Major comments:
The paper under review has serious flaws, mainly in material and methods.
Authors focused only in generic β-glucuronidase positive Escherichia coli according ISO 16649-2 Microbiology of food and animal feeding stuffs (2001), but not to ISO 16649-1:2018 Microbiology of the food chain from 2018. It should be change due to Authors highlights „that consumers may be exposed along the food chain to MDR E. coli, presenting a major hazard to food safety and a risk to public health.
The authors have chosen only one technique to detect β-glucuronidase positive E.coli, but determine pathotypes ETEC, EIEC, EAEC, EPEC, EHEC/STEC, which often are β-glucuronidase negative. β-glucuronidase positive E.coli, is indicator of E.coli/fecal contamination of food, especially meat, but β-glucuronidase activity is not associated with virulence of E. coli.
Line 261 - Sixty fecal samples from breeding (n = 31) and egg laying (n = 29) hens were collected from the Centro and Lisboa and Vale do Tejo regions, between March and May 2019 – from how many farms?
Line 280 – Lack of biochemical or PCR confirmation of E. coli identification.
Line 281 - Phylogenetic grouping and determination of Escherichia coli Pathotypes – lack of positive and negative control. Did the Authors used any process control for the amplifications performed? How are they sure they had not false postives or false negatives?
Line 287 and 292 – Authos used PCR NZYTaq 2x Green Master which contain loading buffer! And next add another 3 µl of loading buffer and water? This is a carelessness in the methodology description. It ust be change.
Line 298 - ETEC, EIEC, EAEC, EPEC, EHEC/STEC – What determine the being of these type of E. coli? Which virulence factor? Lack of positive and negative control.
All PCR condition with PCR size (bp) and references should be added in table.
For antibiotics disk concentrationin µg of antimicrobial agents should be added.
Line 314 - EUCAST [35] – wrong reference it should be [36], like in CLSI [37]. Order of references must be changed. In line 298 and 290 are [51], [52], [53], and next in line 314 [36] and [37] , not citeted before.
Line 330 - Detection of extended-spectrum β-lactamase resistance genes - lack of positive and negative control. Moreover ESBL should be confirmed phenotypically with double dick synergy method (using β-lactam and β-lactamase-inhibitor disks), combined disc method or E-test.
Minor comments:
Line 47 - Enterobacteriaceae
Line 60 - … substantially reduced when cooking is appropriate. – not only by cooking
Line 64 - lack of association of type chosen samples (feces) with egg production, especially with breeding hens, parental flocks for broilers, not producers of consumption eggs.
Line 70 - Campylobacter jejuni, coli, lari. Listeria outbreaks was no due to raw egg contamination, but contamination of hard-boiled eggs. Thus it was no direct association with hens.
Line 74 – 76 - This sentence is confusing, due suggesting using antibiotic not fot treatment but like a type of prevention. It should be rewritten.
Line 134 -137 - This sentence is confusing. Moreover Authors did not mentioned about APEC before. It should be rewritten.
Line 171 - blaCTX-M.
Line 285 - sterile water – change to DNase/RNase-Free Water; added d to the mixture – change to added to pellet
Line 286 - 2.4 µl of which was – add containing DNA. How DNA concentration and purity was measured?
Line 288 – lack of primer producer
Line 294 – visualisation was conducted with using…. It should be added
Line 289 – 0.50 min its 50s?
Line 305 – lack of MH agar plate and antibiotics disc producer
Line 322 - antimicrobial group instead of categories
Line 330 - Detection of extended-spectrum β-lactamase resistance genes – why Authors used different type of polymerase?
Line 336 – in PCR mixture lack of water
Line 337 – 0.50 min its 50s?
Line 340 - visualisation was conducted with using…. It should be added
Author Response
Thank you for the valuable revision and comments on our paper. The updated version was reformulated according to the reviewer's suggestions. The lines number are related to the file with the track changes.
Revisor #3
Major comments:
The paper under review has serious flaws, mainly in material and methods.
Authors focused only in generic β-glucuronidase positive Escherichia coli according ISO 16649-2 Microbiology of food and animal feeding stuffs (2001), but not to ISO 16649-1:2018 Microbiology of the food chain from 2018. It should be change due to Authors highlights „that consumers may be exposed along the food chain to MDR E. coli, presenting a major hazard to food safety and a risk to public health. The authors have chosen only one technique to detect β-glucuronidase positive E.coli, but determine pathotypes ETEC, EIEC, EAEC, EPEC, EHEC/STEC, which often are β-glucuronidase negative. β-glucuronidase positive E.coli, is indicator of E.coli/fecal contamination of food, especially meat, but β-glucuronidase activity is not associated with virulence of E. coli.
A: We hope we have corrected in this revision the errors that have been marked.
In fact, we have enumerated β-glucoronidase positive E. coli so, we have added this information in the article in lines 25, 148 and 375.
We agree that there are pathotypes that are β-glucoronidase negative, the most important is a STEC O157, but we think we have determined the pathotypes as a characterization of the isolated E. coli obtained from a enumeration method. If the objective of the present work would be the presence of pathotypes the methodologies applied would have been detection methods after sample enrichment. Other studies have described the presence of pathotypes using VRBL supplemented with MUG that is an agar that helps in choosing β-glucoronidase positive E. coli.
Line 261 - Sixty fecal samples from breeding (n = 31) and egg laying (n = 29) hens were collected from the Centro and Lisboa and Vale do Tejo regions, between March and May 2019 – from how many farms?
A: The information was added.
Line 280 – Lack of biochemical or PCR confirmation of E. coli identification.
A: According to the method used, colonies that give a typical blue coloration in TBX medium incubated at 44 °C are considered to be E coli colonies, not requiring any additional tests, however, confirmation was made because we were analyzing fecal samples (not food samples) by lactose fermentation and production of indol. This information has been added to the methods.
Line 281 - Phylogenetic grouping and determination of Escherichia coli Pathotypes – lack of positive and negative control. Did the Authors used any process control for the amplifications performed? How are they sure they had not false positives or false negatives?
A: In the PCR we have used positive and negative controls. This information was added in the methodology described
Line 287 and 292 – Authos used PCR NZYTaq 2x Green Master which contain loading buffer! And next add another 3 µl of loading buffer and water? This is a carelessness in the methodology description. It must be change.
A: The authors agreed, the description of the methodology was wrongly described. The description was changed.
Line 298 - ETEC, EIEC, EAEC, EPEC, EHEC/STEC – What determine the being of these type of E. coli? Which virulence factor? Lack of positive and negative control.
A: A new Table was added containing these missing information (Table 11). In lane 486 the controls used are described.
All PCR condition with PCR size (bp) and references should be added in table.
A: All PCR conditions and references were added in a table.
For antibiotics disk concentration in µg of antimicrobial agents should be added.
A: This information has been added.
Line 314 - EUCAST [35] – wrong reference it should be [36], like in CLSI [37]. Order of references must be changed. In line 298 and 290 are [51], [52], [53], and next in line 314 [36] and [37] , not citeted before.
A: The references have been corrected.
Line 330 - Detection of extended-spectrum β-lactamase resistance genes - lack of positive and negative control. Moreover ESBL should be confirmed phenotypically with double dick synergy method (using β-lactam and β-lactamase-inhibitor disks), combined disc method or E-test.
A: Information regarding controls has been added. Regarding the phenotypically confirmation, we agreed, however, only the gene was detected as mentioned in the title. In line 352 we changed the sentence to “In the present study, the blaTemTEM and blaCTX-MCTX-M genes were detected in three E. coli isolates (5%) and could be considered as potential ESBL producers.”
Minor comments:
Line 47 - Enterobacteriaceae
A: Authors agreed. All terms have been corrected.
Line 60 - … substantially reduced when cooking is appropriate. – not only by cooking
A: We agree that is not only cooking that reduces the number of micro-organisms, we have changed to processed
Line 64 - lack of association of type chosen samples (feces) with egg production, especially with breeding hens, parental flocks for broilers, not producers of consumption eggs.
A: The authors agree that an association was missing between the chosen sample and the two types of chickens. The paragraph has been rewritten and information has been added.
Line 70 - Campylobacter jejuni, coli, lari. Listeria outbreaks was no due to raw egg contamination, but contamination of hard-boiled eggs. Thus it was no direct association with hens.
A: Authors agreed. The paragraph was rewritten.
Line 74 – 76 - This sentence is confusing, due suggesting using antibiotic not fot treatment but like a type of prevention. It should be rewritten.
A: Authors agreed. The sentence was rewritten.
Line 134 -137 - This sentence is confusing. Moreover Authors did not mentioned about APEC before. It should be rewritten.
A: Authors agreed. The sentence was rewritten.
Line 171 - blaCTX-M.
A: The change was made.
Line 285 - sterile water – change to DNase/RNase-Free Water; added d to the mixture – change to added to pellet
A: Authors agreed with changing.
Line 286 - 2.4 µl of which was – add containing DNA. How DNA concentration and purity was measured?
A: These measurements were not performed.
Line 288 – lack of primer producer
A: The information was added to the protocol.
Line 294 – visualisation was conducted with using…. It should be added
A: The information was added to the protocol.
Line 289 – 0.50 min its 50s?
A: Yes, change has been made.
Line 305 – lack of MH agar plate and antibiotics disc producer
A: The information was added.
Line 322 - antimicrobial group instead of categories
A: Author agreed with the change of the term.
Line 330 - Detection of extended-spectrum β-lactamase resistance genes – why Authors used different type of polymerase?
A: The method used for the detection of ESBL resistance genes was developed by some colleagues that have recently published (we added the reference) and they used a different polymerase
Line 336 – in PCR mixture lack of water
A: The authors agree with the lack of information. The information has been added.
Line 337 – 0.50 min its 50s?
A: Yes, change has been made
Line 340 - visualisation was conducted with using…. It should be added
A: The information was added to the protocol.
Round 2
Reviewer 1 Report
All the comments were adequately addressed.
Author Response
Thank you for the valuable revision and comments on our paper. The updated version was reformulated according to the reviewer's suggestions.
Reviewer 2 Report
The manuscript has been greatly improved in all its parts.
I have only two clarifications:
In table 2, as in the tables that follow, I would put in brackets the percentage of strains resistant to ampicillin and enrofloxacin.
Furthermore, in the caption of Table 12 delete "and the volume per reaction."
Author Response
Thank you for the valuable revision and comments on our paper. The updated version was reformulated according to the reviewer's suggestions.
In table 2, we put in brackets the percentage of strains resistant to ampicillin and enrofloxacin as suggested.
In tables 3, 4 and 5, when adding the information, in our opinion, it may cause confusion and difficulties to interpret, so we have chosen not to insert this data.
Table 12 "and the volume per reaction." has been deleted
Reviewer 3 Report
I am glad that the authors added new informations and correct methodological errors.I do not understand the reasoning for not biochemical or PCR identification of E. coli, that would have been something easy and would have added value to the report. However, it is not strictly necessary. Otherwise, the authors followed my suggestions and/or answered my questions.
Line 352: in Methods still lak of information about numbers of farms
Line 280 – Lack of biochemical or PCR confirmation of E. coli identification.
Authors focused on blue coloration in TBX medium, lactose fermentation and indol production (line 372).Indole is produced by upwards of 85 species of bacteria and lactose fermentation is very popular
Some of E. coli "inactive" especially in fecal samples are lactose negative and not producing indole, but still blue in TBX medium. In feces Escherichia hermanni, Citrobacter braakii is also lactose and indole positive. In fecal samples, (not food samples) with rich gut microbiota additional biochemical or PCR tests are required.
Line 476 - names of genes bla should be italic
Author Response
Thank you for the valuable revision and comments on our paper. The updated version was reformulated according to the reviewer's suggestions.
Line 352: in Methods still lak of information about numbers of farms
A: The information has been added as suggested.
Line 280 – Lack of biochemical or PCR confirmation of E. coli identification.
Authors focused on blue coloration in TBX medium, lactose fermentation and indol production (line 372). Indole is produced by upwards of 85 species of bacteria and lactose fermentation is very popular Some of E. coli "inactive" especially in fecal samples are lactose negative and not producing indole, but still blue in TBX medium. In feces Escherichia hermanni, Citrobacter braakii is also lactose and indole positive. In fecal samples, (not food samples) with rich gut microbiota additional biochemical or PCR tests are required.
A: We agree with your comment that more biochemical tests could be performed to identify E. coli. However, the isolation medium differentiates E. coli from other bacteria based in the ß-glucoronidase and although this enzyme can be present in other bacteria such Citrobacter, Enterobacter, Klebsiella, Salmonella, Shigella et Yersinia, its presence concerns only a small number of strains in each of the species mentioned. All the blue colonies chosen were lactose positive and indol positive.
Line 476 - names of genes bla should be italic
A: The names of genes were formatted.